# Building and experimenting with an agent-based model to study the population-level impact of CommunityRx, a clinic-based community resource referral intervention

Stacy Tessler Lindau[1,2,3,4]*, Jennifer A. Makelarski[1‡], Chaitanya Kaligotla[5,6‡], Emily M. Abramsohn[1], David G. Beiser[7], Chiahung Chou[8], Nicholson Collier[5,9], Elbert S. Huang[10], Charles M. Macal[5,9], Jonathan Ozik[5,9], Elizabeth L. Tung[10]

1 Department of Obstetrics and Gynecology, University of Chicago, Chicago, Illinois, United States of America, 2 Comprehensive Cancer Center, University of Chicago, Chicago, Illinois, United States of America, 3 Department of Medicine, Section of Geriatrics & Palliative Medicine, University of Chicago, Chicago, Illinois, United States of America, 4 Bucksbaum Institute for Clinical Excellence, University of Chicago, Chicago, Illinois, United States of America, 5 Decision and Infrastructure Sciences Division, Argonne National Laboratory, Lemont, Illinois, United States of America, 6 Beedie School of Business, Simon Fraser University, Burnaby, British Columbia, Canada, 7 Section of Emergency Medicine, Department of Medicine, University of Chicago, Chicago, Illinois, United States of America, 8 Department of Health Outcomes Research and Policy, Auburn University, Auburn, Alabama, United States of America, 9 Consortium for Advanced Science and Engineering, University of Chicago, Chicago, Illinois, United States of America, 10 Section of General Internal Medicine, Department of Medicine, University of Chicago, Chicago, Illinois, United States of America

‡ Co-second authors
* slindau@bsd.uchicago.edu

**Data Availability Statement:** The model code and the workflow code used to implement the

## Abstract

CommunityRx (CRx), an information technology intervention, provides patients with a personalized list of healthful community resources (HealtheRx). In repeated clinical studies, nearly half of those who received clinical "doses" of the HealtheRx shared their information with others ("social doses"). Clinical trial design cannot fully capture the impact of information diffusion, which can act as a force multiplier for the intervention. Furthermore, experimentation is needed to understand how intervention delivery can optimize social spread under varying circumstances. To study information diffusion from CRx under varying conditions, we built an agent-based model (ABM). This study describes the model building process and illustrates how an ABM provides insight about information diffusion through *in silico* experimentation. To build the ABM, we constructed a synthetic population ("agents") using publicly-available data sources. Using clinical trial data, we developed empirically-informed processes simulating agent activities, resource knowledge evolution and information sharing. Using RepastHPC and chiSIM software, we replicated the intervention *in silico*, simulated information diffusion processes, and generated emergent information diffusion networks. The CRx ABM was calibrated using empirical data to replicate the CRx intervention *in silico*. We used the ABM to quantify information spread via social versus clinical dosing then conducted information diffusion experiments, comparing the social dosing effect of the intervention when delivered by physicians, nurses or clinical clerks. The synthetic

parameter space characterization experiments are publicly available at https://github.com/jozik/community-rx).

**Funding:** Research reported in this publication was supported by the National Institute on Aging of the National Institutes of Health R01AG047869 (S.T.L, J.A.M., E.M.A., D.G.B., C.C., N.C., E.S.H., C.K., C.M.M., J.O.), R01AG064949 (S.T.L, J.A.M., E.M.A., E.S.H.), and K24AG069080 (E.S.H.), the National Institute of Minority Health and Health Disparities R01MD012630 (S.T.L., J.A.M., E.M.A.), the National Institute of Diabetes and Digestive and Kidney Diseases P30DK092949 (E.S.H.) and the National Heart, Lung and Blood Institute K23HL145090-01 (E.T.). The full amount of the project costs were financed with federal money. The funders had no role in study design, data collection and analysis, decision to publish, or preparation of the manuscript. The content is solely the responsibility of the authors and does not necessarily represent the official views of the National Institutes of Health.

**Competing interests:** I have read the journal's policy and the authors of this manuscript have the following competing interests: Under the terms of prior Department of Health and Human Services, Centers for Medicare & Medicaid Services funding (1C1CMS330997), innovators were expected to develop a sustainable business model to continue and support the model that was tested after award funding ended. Dr. Stacy Lindau is the founder and Chief Innovation Office of NowPow, LLC, a Unite Us company. She is also founder and president of MAPSCorps, 501c3. Neither the University of Chicago nor University of Chicago Medicine is endorsing or promoting any NowPow/MAPSCorps/Unite Us Entity or its business, products, or services. No other authors of this manuscript have competing interests.

population (N = 802,191) exhibited diverse behavioral characteristics, including activity and knowledge evolution patterns. *In silico* delivery of the intervention was replicated with high fidelity. Large-scale information diffusion networks emerged among agents exchanging resource information. Varying the propensity for information exchange resulted in networks with different topological characteristics. Community resource information spread via social dosing was nearly 4 fold that from clinical dosing alone and did not vary by delivery mode. This study, using CRx as an example, demonstrates the process of building and experimenting with an ABM to study information diffusion from, and the population-level impact of, a clinical information-based intervention. While the focus of the CRx ABM is to recreate the CRx intervention *in silico*, the general process of model building, and computational experimentation presented is generalizable to other large-scale ABMs of information diffusion.

## Author summary

CommunityRx (CRx) is a clinic-based intervention that provides patients with information about community resources for health-maintenance and promotion. Prior work found that nearly half of people exposed to CRx share their resource information with others. This study describes construction of and experimentation with an agent-based model (ABM) to examine the potential impact of CRx and other health information interventions on the broader community via social spread or "dosing" from people directly exposed to the intervention. We show how we integrated clinical trial, demographic and epidemiologic data and expert informant insights to develop and assign behaviors to a synthetic study population (agents). Using CRx clinical trial data, we then delivered the intervention to these agents and simulated information spread. We describe *in silico* experimentation to illustrate insights about information spread generated by the ABM that complement clinical trial findings. This study shows how data from individual-level clinical and population studies can be used to create a computational laboratory to assess the broader impact of a health information intervention. In addition to inspiring integration of individual-level and systems science approaches to the study of health information interventions, this study enables peer review to inform model iteration and experimentation.

## Introduction

Over the last five years, with the shift to value-based care, community resource referral technologies have been adopted by many U.S. health systems.[1–3] These platforms, which aim to connect patients to community-based resources for health-related socioeconomic needs (e.g. food and housing support, transportation) serve to advance health systems' population health care delivery models and their efforts to mitigate inequities due to social and structural determinants of health and disease.[4,5] Mainstream data sources about healthful community resources are commonly outdated, idiosyncratic, analog, lack eligibility information and must be accessed outside the usual clinical informatics workflow.[3,6] As a result, clinician and patient efforts to make use of these resources to promote health or manage with illness are often frustrating and inefficient.[3] Sustainability of these interventions in healthcare practice depends not only on their seamless integration with usual clinical workflows, but also on their demonstrated impact on individual and population health-related outcomes and costs.

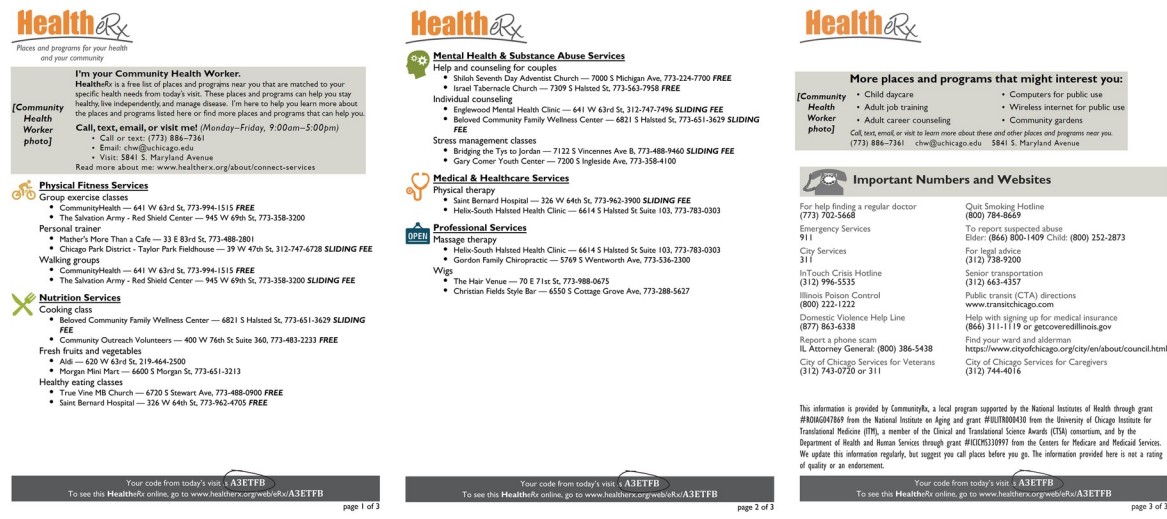

**Fig 1. Sample HealtheRx generated for a patient with cancer.**

In contrast to medical drug and device interventions, information interventions delivered by a clinician to a patient can quickly spread to other community members through social networks. Evidence suggests that social networks can potentiate the impact or credibility of health-related information [7,8] and that patients generally view information obtained from a clinician as highly trustworthy.[9] Therefore, information spread through social networks from an intervention delivered in the clinical setting has the intriguing potential to meaningfully impact people in the community beyond those who directly received the intervention. Alternatively, negative information or misinformation spread could be harmful to individual and public health efforts. However, traditional clinical trial designs, which typically assess the impact of an intervention on the individual patient, do not capture the dynamic, multi-level impact of information-based health interventions.

CommunityRx is a scalable, evidence-based information technology intervention designed together with a broad diversity of stakeholders, including residents, on Chicago's South Side to promote population health by connecting patients to health-promoting community resources ("resources").[10,11] CommunityRx was created, in part, due to systematic deficiencies in mainstream data sources which under-represent health-promoting businesses and organizations in higher poverty communities.[6] These deficiencies impede the efforts of healthcare professionals to meaningfully and equitably execute on clinical best practice guidelines that indicate a wide range of community resources for various health conditions.

The primary mode by which CommunityRx disseminates resource information is via the "HealtheRx," a printed list of local resources personalized to the patient's age, gender, home address, health conditions and preferred language (Fig 1). A HealtheRx is auto-generated at the point of medical care using software algorithms integrated with the electronic medical record (EMR) to match individuals to an indicated set of community resources (Fig 2). With the press of a digital button in the EMR by a clinician (e.g. physician, nurse or clerk), extant data in the patient's chart (e.g., age, gender, home address, preferred language, ICD-9/10 codes) are consumed into the CommunityRx algorithm to generate a HealtheRx.

Indicated resources for each condition and status are informed by best clinical practice guidelines, expert opinion and community member input about self-care activities for more than 30 common health conditions (e.g., diabetes, hypertension, obesity), social conditions (e.g. food insecurity, housing instability, domestic violence) and statuses (e.g., newborn,

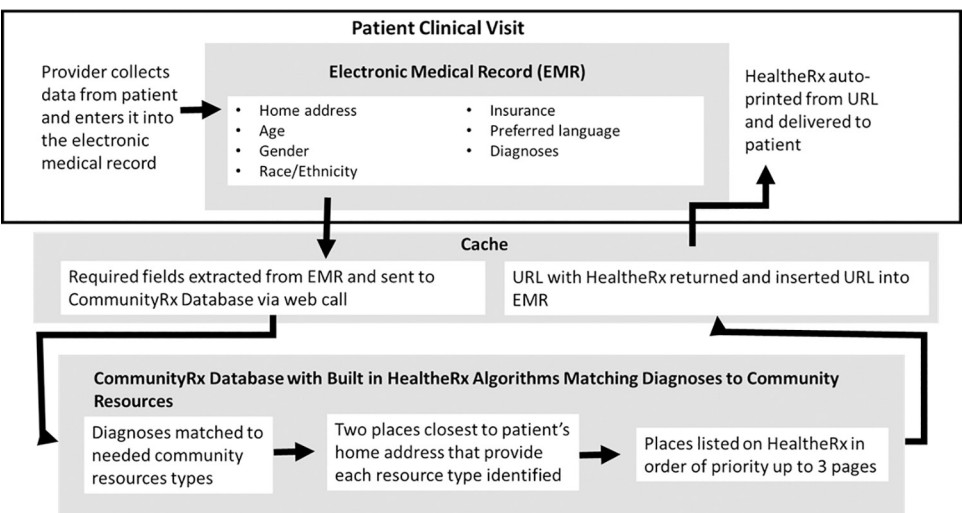

**Fig 2. Visualization of how a HealtheRx is Generated Using Data from a Patient Electronic Medical Record and the CommunityRx Software Algorithm.**

adolescent, pregnant). Each HealtheRx includes resources to address an individual's basic needs (e.g. food, housing), physical and mental wellness (e.g. fitness, nutrition, yoga), disease self-management (e.g. smoking cessation, weight loss) and caregiving (e.g. cancer support, respite care, home hospice) needs. Clinicians deliver the HealtheRx to patients at discharge from a medical visit, including emergency and hospital care. An independent quasi-experimental study of CRx comparing people who received one or more HealtheRx to matched controls found a significant decrease in hospitalization among Medicare beneficiaries and a decrease in emergency department utilization among Medicaid beneficiaries.[10,12,13] Cost savings over the short term were modest to neutral. Delivery of the intervention by a clerk is less costly than delivery by a nurse or physician. Experimentation is needed to inform the lowest cost delivery method while maintaining information spread to the broader population.

Inspired by the Kilbridge et al. prescription management process for drugs, community resource prescribing requires several steps.[11,14] The community resource prescribing "decision-making" process is initiated during the clinical encounter, supported by the software algorithms described above. The personalized prescription is auto-generated from extant data in the patient's EMR (e.g. age, gender, home address, preferred language, problem list) plus any new data (e.g. a new diagnosis or social risk) entered during the visit. Once the HealtheRx is generated and printed, it is "fulfilled." Fulfillment begins with "evaluation" when a clinician or clinical staff member reviews the HealtheRx with the patient and is "dispensed" when the patient contacts or visits the community resource provider. The community resource is "administered" when the patient uses, receives or consumes the recommended resources. Kilbridge's drug e-prescribing model does not contemplate the possibility of sharing or diffusion (although prescription medications are sometimes shared).[15] However, in the CRx iteration of the e-prescribing process model, we did anticipate that prescribed information could be shared. We therefore designed our primary research to assess for sharing.

With initial funding from a Center for Medicare and Medicaid Innovation award (3/2013-5/2015, "CRx-1"), CRx generated 253,479 HealtheRxs for more than 113,000 patients over a 30 month period.[11] The HealtheRxs were typically delivered by a physician, nurse or clinical clerk, depending on local site preferences. All patients living in the target geography were

eligible, including tens of thousands of people with cancer, cardiovascular disease, pulmonary disease, obesity and other common chronic conditions. In a telephone survey of 458 HealtheRx recipients ages 18–93 years enrolled in CRx-1, 71% reported that they learned about a new resource from the HealtheRx and 83% reported the HealtheRx was "very useful." In a subsequent real-world clinical trial (CRx-2), 411 patients ages 45–74 years were followed for three months after receiving a HealtheRx. Key outcomes, including self-efficacy for finding and knowledge about healthful community resources improved.[10,16] In addition, the CRx-1 and 2 studies yielded evidence of both intervention and information spread. Soon after the intervention was deployed at the first community health center, a physician "hacked" the CRx system in order to be able to use the intervention at a clinic site where he worked outside the target geography, thereby causing unexpected, but desirable spread of the intervention to other communities.[11] In CRx-1, 49% of participants reported sharing resource information from their HealtheRx with at least one person;[11] this finding was replicated in CRx-2.[10] Additionally, about half of nurses and physicians who participated in CRx-2 also shared HealtheRx information with others.[17] The repeated finding that information from the clinical intervention spread beyond the HealtheRx recipients to a more general population motivated a systems science approach to evaluating the impact of CRx.

Systems science methods are increasingly being used to complement traditional epidemiologic and ecological studies,[18] although few clinical trials have incorporated a systems science approach. Observational and experimental studies can occur in tandem with, and inform, the building of computational models.[19–21] Agent-based modeling (ABM) is a systems science simulation methodology that captures emergent behavior among individuals ("agents") as a function of interactions between agents and their environment. ABM is particularly useful for studying phenomena affected by second-order effects resulting from stochastic and network interactions (e.g. the population-level effect of an individual-level intervention where individuals interact with and affect each other's behavior).[22] Additionally, ABM enables linkage of individual choice behavior to emergent population-level outcomes (like total clinic visits or other health maintenance activities). In contrast to other methods (e.g. differential equations modeling) these micro-to-macro level dynamics make ABMs particularly well suited to study an informational intervention. ABM can also serve as an efficient adjunct for experimentation that is outside the limited scope of a clinical trial or too time-consuming or otherwise costly to conduct *in vivo*.

ABM has been used to examine the spread of infectious disease[23–25], effects of environmental exposures [26,27] and health-related interventions and policies.[28–36] Barbrook-Johnson et al. used ABM to examine the effects of the TELL ME intervention, an information-based intervention that deployed public health communication strategies to reduce the spread of influenza.[36] The TELL ME ABM was designed to predict the effect of communication strategies on influenza spread, but did not run simulations specifically to model information flow and dynamics. A description of the validated TELL ME ABM was presented as a teaching tool, to demonstrate how such models can be used to inform policy, even in the absence of complete data. Best practice dictates that the first step in any systems modeling development is to create the proper model design and then subject the design to evaluation including peer review.[37] We have previously described in the systems science literature the technical design details of the ABM used in this study as well as the associated model validation.[38,39]

Informed by and extending this work, the goals of this study are to: (1) describe the interdisciplinary method of model building and data sources used to develop an ABM to study an information-based health intervention, and (2) describe an *in silico* experiment to illustrate how an ABM can generate insights to complement a clinical trial of an information-based intervention. First, using the development of the CRx ABM as a case study, we describe

interdisciplinary model building methods that could generalize to other information-based healthcare interventions. Next, to illustrate how ABM experimentation can complement findings generated using traditional clinical trial methods, we describe an *in silico* experiment to deepen our understanding of the information sharing findings from related clinical trials. To our knowledge, this is the first study to use ABM to examine the population-level information diffusion dynamics resulting from a clinical information intervention.

## Methods

### 1.0 Rationale for building the CommunityRx ABM

The CRx clinical trials provide first order evidence of information spread from the people exposed to the intervention (patients and clinicians) to others in their social networks. The CRx ABM was designed to simulate and experimentally probe, *in silico*, the flow and spread of this information at scale, across the larger population under varying conditions. Spread of community resource information from patients to others in the community is a desirable dynamic that could serve as a force multiplier for CRx and other information-based health interventions. Although we saw no empirical evidence of negative information or misinformation spread in prior studies, understanding the flow of negative or misinformation is also of interest in future instances of the model. Here, we describe the interdisciplinary model-building process and one intervention delivery experiment conducted *in silico* to inform future iterations and implementation of the intervention.

### 2.0 Interdisciplinary model-building process

The interdisciplinary process for building the CRx ABM involved several steps, informed by extant literature[21] and illustrated in Fig 3. Investigators, including physician scientists from the University of Chicago's Biological Sciences Division and Consortium for Advances Science and Engineering, and Argonne National Laboratory's Decision and Infrastructure Sciences Divsion, met regularly over the course of several years to: (1) come to a shared understanding about and vocabulary to enable interdisciplinary study of the CRx intervention, (2) identify and integrate public use, clinical trial and other data sources with the ABM drawing on complementary expertise (e.g. the systems scientists were experienced with time use data and the others were experienced with public health and trial data), (3) fill data gaps by designing and implementing new data collection strategies drawing on qualitative research expertise of both groups and survey research skills of the biomedical scientists, (4) describe and validate the model,[39] and (5) design and implement *in silico* experimentation (a focus of this paper).

In addition to observational data from publicly available demographic, economic and epidemiologic datasets[10,11,40,41] and primary data collection from clinical trials, the CRx ABM also used data sourced from expert informants. Expert informants—referred to as subject matter experts in the systems science literature—are commonly used in agent-based modeling to fill gaps in extant data.[42] Among the expert informants involved in developing the CRx ABM were people who lived in, shared demographic characteristics with and had decades of experience serving and providing medical care for people in the target geography. Primary data collection, including two clinical trials (CRx-1 and CRx-2), occurred in parallel with the CRx ABM building, with each approach informing the other (Fig 3).

### 3.0 Overview of the CRx Agent-Based Model

Here, we provide an overview of the CRx ABM with an emphasis on how the ABM was built, using the Grimm et. al. Overview, Design concepts and Details (ODD) summary protocol.[43]

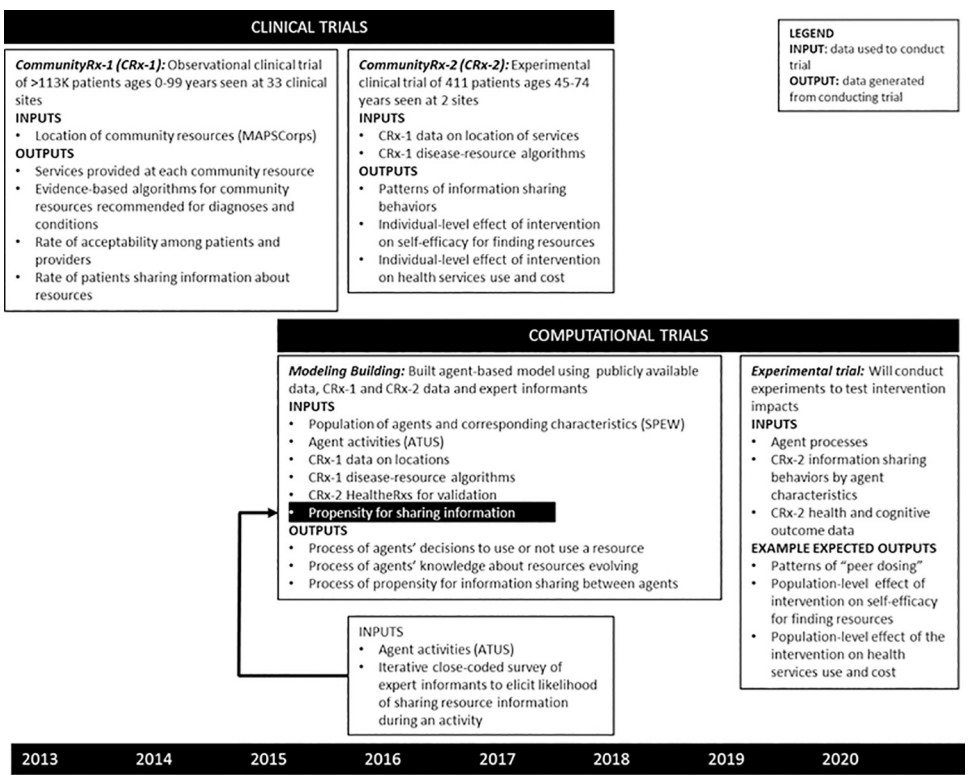

**Fig 3. Interdisciplinary process and timeline of clinical and computational trial activities with data inputs and outputs.**

Comprehensive description of the model design, calibration and validation, including all of the components indicated for a full ODD, have been previously published.[38,39] Language used verbatim from the Grimm et. al. summary ODD template is italicized and bold in the sections below and many section headers derive directly from the Grimm protocol. The model and workflow code used to implement the parameter space characterization experiments are publicly available at https://github.com/jozik/community-rx. A description of model processes that occur in every timestep is provided in Appendix A in S1 Text. The CRx ABM was implemented in C++ using the Repast for High-Performance Computing ABM toolkit and the Chicago Social Interaction Model (chiSIM) framework.[44–47] The CRx studies were approved by the University of Chicago Institutional Review Board.

## 3.1 CRx ABM Purpose and Questions

***The overall purpose*** of the CRx ABM is to (a) demonstrate the flow and spread of resource info from primary agents (those who received one or more clinical doses or HealtheRxs) to others in the community and (b) conduct experiments on how delivery of the HealtheRx and other variable conditions can impact the flow and spread of information to the community. ***Specifically, we are addressing the following questions:*** (1) Can we recreate, *in silico*, delivery of the CRx intervention, including generation of personalized HealtheRxs ("clinical dosing")? (2) Can we recreate the underlying dynamics of resource information diffusion ("social dosing") following *in silico* delivery of an intervention delivered to primary agents at the point of clinical care? (3) How does variation in clinical parameters (e.g. who delivers the HealtheRx to the primary agent) affect information diffusion to the broader population?

### 3.2 CommunityRx ABM Patterns

To establish that we have created a working computational laboratory for experimentation, the CRx ABM building process involved a systematic examination of *patterns* at the individual and system level that are salient to the purpose of the model. ***To consider our model realistic enough for its purpose,*** we used the following criteria: (a) the demographic characteristics of the synthetic population would reflect those of the actual population, (b) the synthetic population would exhibit diverse behavioral characteristics, including agent activity and knowledge evolution patterns, (c) *in silico* delivery of the intervention would be replicated with high fidelity, (d) large-scale information diffusion networks would emerge among agents exchanging resource information, (e) varying the propensity for information exchange would result in networks with different topological characteristics. Examination of these patterns is presented in the Results section of this paper.

### 3.3 CommunityRx ABM Entities

In general, the main components of a population-based ABM include model entities and behaviors. Entities include a synthetic population of agents who statistically correspond to the population in the study area and an environment, which typically refers to a physical environment. Behaviors include agent behaviors and a set of interactions for each agent, such as receiving or sharing information.[42] The CRx ABM ***includes the following entities***–a synthetic population of agents and a geographical environment that together are statistically representative of the community under study (16 ZIP codes on Chicago's South Side, a predominantly African American/Black demographic where the CRx intervention was created and studied). ***The state variables characterizing these entities*** are listed in Table A in S1 Text.

Using the publicly available Synthetic Populations and Ecosystems of the World (SPEW) 2016 dataset (http://stat.cmu.edu/~spew/resources/),[40] we built a synthetic population of people (agents) ages 16 years and older (the HealtheRx for people younger than 16 was typically given to an accompanying adult). The individual characteristics of the synthetic population and their distribution are presented in the Results section and Table B in S1 Text. Population characteristics were static.

Building the environment involved imbuing the model with the physical locations ("places") that agents could occupy within the 16 ZIP code (106 mi$^2$) CRx study. This region comprises a little less than half of Chicago's geography. The physical environment included: (1) households, workplaces and schools (data for these locations were also obtained from the SPEW dataset); (2) healthcare sites (clinics) where agents could receive a HealtheRx, and (3) health-promoting community-based organizations and business (places) where agents could go to use services ("resources" are specific places providing specific services [e.g., fitness, weight management, smoking cessation]) prescribed on the HealtheRx.

Agents could also co-locate at these places and exchange information about resources with other agents. The locations of healthcare sites and places where resources were administered (N = 4,903) were obtained from a 9/16/2016 snapshot of the dynamic CRx resource inventory. The CRx resource inventory was built using two primary data sources: (1) MAPSCorps (http://mapscorps.org), a dataset created by an organization that trains local youth who conduct an annual door to door survey of the name and location of every place providing services in the target region[6,48] and (2) community health workers who conducted regular phone-based surveys with places to capture the types of services provided at each location.[11,49]

Primary model behaviors include (a) processes of information sharing (how information flows and spreads or diffuses from an individual agent to others in the community) and (b)

decision behaviors (how agents choose to utilize resources as a function of information received). These processes and behaviors are described below under "Important Processes."

## 3.4 CommunityRx ABM spatial and temporal resolution

The CRx ABM is spatially explicit. Each place in the synthetic environment, including health-promoting resources, are provided at an actual physical place with a fixed latitude and longitude in the 16 ZIP code study region. Each time step in the model represents one hour of simulated time and simulations are run for a period of at least 4 weeks. We generally observed stable behavior by week 3 of the simulation. Agents are assigned specific activity schedules for every 24 hours, representing a simulated day. The extent of the CRx ABM is in recreating the CRx intervention within the defined spatial and temporal resolution.

Structurally, the CRx ABM is a time-stepped activity simulation in which agent states are updated at each simulation time interval (hourly), based on agent behaviors and interactions. Using a demographically matched, randomly selected daily activity schedule, the simulation component of the model determines an agent's activity for the current time step and moves each agent to the activity's location. Co-located agents share information about specific community resources based on the activity being performed. Based on information received and retained through this process, each agent decides whether or not to perform certain health-promoting activities (a subset of all possible activities) utilizing community-based resources that were prescribed to the agent by the CRx intervention. Thus, we can derive population-level metrics of resource information diffusion. Further, as a function of information diffusion and agent decisions about health-promoting activities, we can also derive population-level metrics of resource use. A description of model processes that occur in every time step is provided in the following subsection and summarized in paragraph 3 of Appendix A in S1 Text.

## 3.5 The most important model processes at every time step

**3.5.1 Agent activity behavior.**   In addition to sociodemographic characteristics, agents were imbued with activity schedules for each simulated day. Agent activity schedules were developed from the publicly available 2016 American Time Use Survey (ATUS) dataset (https://www.bls.gov/tus/data.htm).[40] This dataset included 10,493 activity schedules covering weekdays and weekends, each associated with the demographic characteristics of survey respondents. By matching the SPEW and ATUS datasets on age, gender, race and ethnicity, each agent in the synthetic population was assigned a daily activity schedule, in 1-hour increments, from a distinct set of available ATUS schedules for weekdays and for weekends.

This matching resulted in a set of available schedules for each agent ranging between 2 to 1496. For each simulated day, an agent was randomly assigned a unique activity schedule from the set of available matched schedules. These daily schedule assignments resulted in agents being co-located in unique places. Information sharing occurred when co-located individuals interacted. The large range of available schedules for the daily random assignment was a modeling choice to introduce randomness of available activity schedules for each demographic subgroup in our agent population (alternatively, agents would follow the same schedules every simulated day). While a larger sample of available schedules does indeed introduce stochasticity, it was deemed better than the case of having groups of agents with very few schedules, which could result in unrealistically repetitive activity patterns. Future work will focus on the implication of this randomization on the resulting co-location networks. We also do not account for the seasonality of schedules due to the lack of available granular data. The implications of these modeling choices are a limitation of our model and the focus of future work.

Using the ATUS data, we classified all activities into one of two subsets: (1) activities of daily living (e.g., sleeping, washing and grooming, talking on the phone) or (2) physical and mental health maintenance or promotion ("health maintenance") activities at both the individual (e.g., health-related self-care, doing yoga) and household level (e.g., obtaining medical care for a child or an adult in the household). Applying an iterative data collection and discussion process with expert informants, we then compared the subset of health maintenance ATUS activities to the complete list of service types listed on a HealtheRx ("service types" refers to kinds of services [e.g., fitness, weight management, smoking cessation] that a place could offer in contrast to a "resource" which refers to a specific place providing a specific service). Each ATUS activity from this subset was mapped to one or more relevant HealtheRx places where an agent might reasonably carry out that ATUS activity. For example, the ATUS activity "providing medical care to household children" was mapped to the HealtheRx services "fill prescriptions," "medical supplies," and "home care." Expert informants were presented with both the subset of ATUS activities and HealtheRx service types and discussed the likelihood of an agent using a service while performing an activity for each mapping until consensus was achieved. Using the CRx resource inventory, each service type was associated with all places in the physical environment that provided that service.

We computed the average number of minutes per week spent doing each activity for each agent across all activity schedules that were demographically matched to that agent. Next, the time per week during which a given HealtheRx service could be used was determined by summing each agent's average time doing every ATUS activity mapped to that HealtheRx service and dividing by the total number of agents for each matched schedule. This average was computed within each of four age groups (<31, 31–45, 46–65, and 65+ years), weighted by the number of people in the age group (Fig 4). These groups were defined using demographic, life course and healthcare policy considerations and are consistent with commonly applied strata for age in the clinical and population health literature.

**3.5.2 Agent knowledge evolution.** Agents maintained dynamic β scores for resources they knew about. At initialization, agents were seeded with knowledge about resources using a distance-based random assignment algorithm. Agents were seeded with knowledge of 10 to 100 resources located within a low distance radius (parameterized as 1 mile), 1 to 5 resources within a medium distance (1–3 miles) and 1 to 5 resources for long distances (3+ miles). The first two weeks were considered a burn-in period for the entire simulation. We generally observed stabilized behavior by week 3 of the simulation. Results reported in this paper use output from week 3 of the simulation.

The β scores were boosted by information dosing, a factor based on exposure to resource information and the source of the information. Agents could be exposed to resource information by doctors, nurses, clinical clerks, social contacts and by first-hand use of the resource. The clinician sources provided information to agents by delivering a HealtheRx during a healthcare visit at a clinic ("clinical dose"). Social dosing occurred when agents co-located at places (e.g., a barber shop) and exchanged resource information (e.g. about the gym). Agents used a resource by going to its location (subsequently referred to as just "use"). The functional form of β score evolution and its parameterization were created in consultation with expert informants and through sensitivity analyses as previously described.[39] The β score for agent $i$ about resource $j$ at time $t$ evolved according to the following functional form and is described below:

$$\beta_{i,j}^t = \lambda (\beta_{i,j}^{t-1})^{\varepsilon_x(1-\beta_{i,j}^{t-1})+\beta_{i,j}^{t-1}} \tag{1}$$

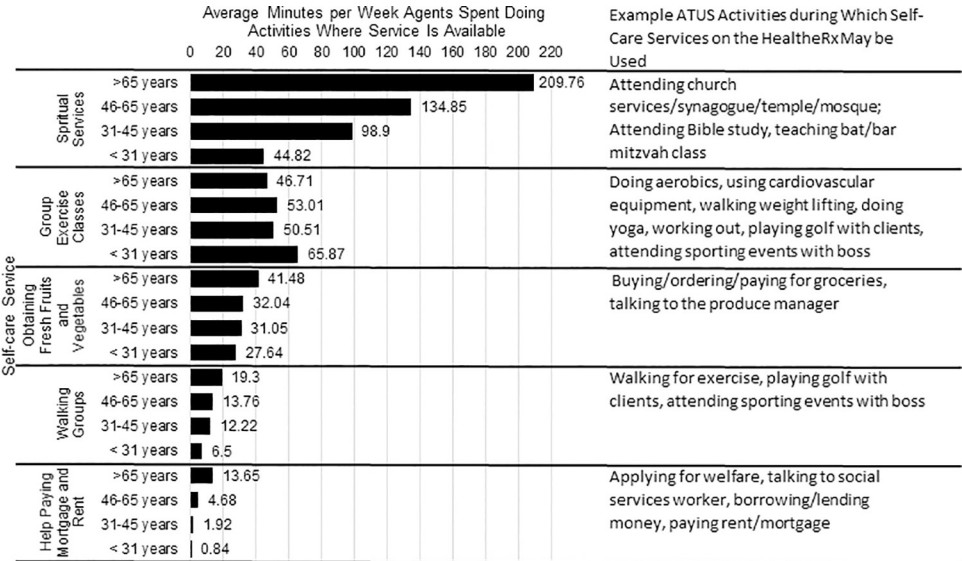

**Fig 4. Average minutes per week, stratified by age agents spent doing activities during which they could use a self-care service listed on the HealtheRx; Chicago, Illinois 2016–2018.**

The decay parameter λ (0.991 for the simulation run producing Fig 5, with an allowable range of values between 0.991 and 0.9994) accounted for receding knowledge of a resource. The source of resource information was associated with a value, $\varepsilon_x$, where $x \in$ (*Doctor, Nurse, Clinical Staff, Use, Peer, None*}. "Use" indicated an agent had been to or consumed a given resource following the decision function in Eq 1.

For the simulation run used to produce Fig 5, the following values were used $\varepsilon_x$ = {0.05, 0.15, 0.25, 0.2, 0.9, 1}. Sources expected to have a greater influence on the agent had higher dosing values (corresponding to smaller $\varepsilon_x$ values). As modeled, $\varepsilon_x$ only accounts for the

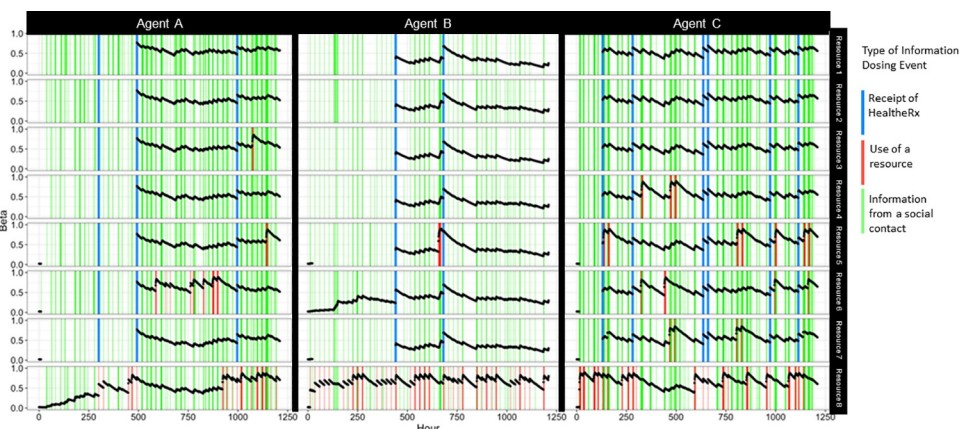

**Fig 5. Exemplar instance of the evolution of three agents' knowledge (Beta, β) of eight resources over time (λ of 0.991 used in this instance based on sensitivity analysis and model calibration previously reported in [39]); Chicago, IL 2016–2018.** Note: Each column (n = 3) represents a unique agent. Each row represents a unique resource (n = 8). Each black dot indicates the β scores (left y-axis) at in point in time in hours (bottom x-axis). Information dosing events (receipt of information about a given resource) that occurred during a given hour are indicated by vertical lines as: receipt of a HealtheRx (blue), receipt of information about resources from a social contact (green) and use of a resource (red).

source from which the information was received and not the type of resource information received. This modeling choice was based on one of our specific questions: how does the information source influence information diffusion dynamics? This question is salient to real-world deliberations about how best to deliver community resource referrals in the clinical context. The ABM is designed so it could easily be iterated to include a resource-specific information dynamic. Furthermore, information dynamics may vary by factors like the stigma associated with information about a given resource type (e.g. HIV/AIDS support group, food pantry or substance use counseling). Future model development could be iterated to account for this kind of dynamic. As referenced previously, the knowledge evolution process does not account for a negative effect (where negative or incorrect information could lead to an agent avoiding a specific resource). Although related studies have surfaced no evidence for negative information spread, it is certainly possible. This factor is a limitation of the model left for future model development.

Agents maintained knowledge about a maximum of 200 resources; resources with lower β scores were replaced by resources with higher β scores over time. The assumption of a memory load of 200 resources was made to represent the bounded knowledge and ability of the agents to maintain information for their decision-making (described below). The CRx ABM allows for the size and heterogeneity of agents' cognitive capacity to be parameterized. Through the dynamics of agents' β scores, the CRx ABM models internal agent states with respect to resource knowledge.

**3.5.3 Agent information sharing behavior.** Information sharing was also modeled to depend on the nature of individuals' activities (e.g., there is no information exchange when an agent was sleeping, whereas information can be exchanged when co-located agents were exercising in a gym). The propensity for an agent to receive information ("p-score") was dependent on the activity in which the agent was engaged. Each ATUS activity in our subsets was assigned a propensity for information sharing. Information sharing was assumed to occur between two or more co-located people by way of face-to-face conversation. (Therefore, information shared digitally, for example, was not captured in the model.)

Because we found no extant data to inform assumptions about propensity for information sharing during various activities, we surveyed expert informants in iterative rounds to generate p-scores for each relevant activity. The survey elicited the likelihood (p-score = none, low, medium, high) that the respondent would receive resource information from another person while doing a given activity. For example, "sleeping" uniformly generated a p-score of "none." Activities like "helping household adults" and "grocery shopping" produced a p-score of "medium" and "socializing, relaxing, and leisure as part of job" and "obtaining medical and care services for household adult" produced scores of "high." Agents at a location share information with other co-located agents based on a threshold defined by the propensity for resource information sharing during that activity and individual random draws against that threshold. The characteristics of agents receiving information were not a factor in the information sharing dynamic. The design of this mechanism was chosen to reflect the fundamental information sharing dynamic commensurate with the expected propensity of resource information sharing determined with expert opinion, and can be considered as a limitation imposed by the data on the model. However, this information sharing dynamic does allow us to compare social dosing effect to the direct clinical dosing effect. The parameter values for these effects were critical to model calibration and was previously described in detail.[39]

**3.5.4 Agent decision-making.** We then developed processes for agents' decisions about using a resource (recall, defined as a specific service at a specific place). These decisions were dependent on agents' information-sharing behavior and their knowledge evolution. Each of the parameters involved in the decision processes was informed by empirical data and expert

opinion (Table A in S1 Text). Sensitivity analyses performed on selected parameters have been previously described.[39]

The agent's decision to use a resource was modeled as a binary choice A/B Decision Model. Each activity was characterized by a decision type–A/B decision or not. Activities classified as health maintenance or promotion activities (the activities that the CRx intervention targets) faced an A/B decision choice. Other activities not related to health-maintenance behaviors did not require a decision choice–agents would proceed with such activity at the designated location. Agents presented with an ATUS activity in their daily schedule that was subject to an A/B decision faced a choice to use a resource or not. For example, when presented with the ATUS activity "fitness," the agent chose whether to use a fitness facility (*Decision A*) or not (*Decision B*). If the agent decided to use the resource, they proceeded with the activity in their schedule mapped to that service. If they chose not to use the resource, they continued doing the previous activity in their schedule.

The following agent characteristics dictated the decision process for each agent $i \in P$, *where P is the set of all agents*:

i.  natural activation level ($\alpha_i$), derived from Skolasky et al. [50]

ii. a function of their knowledge ($\beta_{i,j}$) about a specific resource $j \in R$, *where R is set of all resources* (e.g., a community center), where they could use a specific service (e.g. a gym),

iii. the distance to the resource for the agent ($\delta_{i,j}$), and

iv. the inherent inertia that they needed to overcome for using that resource ($\gamma_j$).

An agent's decision to use the resource, *Decision A*, was made when the agent's activation level (threshold) was exceeded by the agent's combination of knowledge and effort required (distance and inertia). The relationship at the time (t) is given by:

$$if \ \frac{\beta_{i,j}^t}{\delta_{i,j}\gamma_j} > \alpha_i, \ then \ Decision \ A, \ else \ Decision \ B \tag{2}$$

To provide an illustrative example–consider an agent following their daily hourly schedule until they come to an activity requiring an A/B decision: go to a gym or not. The agent uses the decision calculus described above to choose between the gym (A) or their current (B) activity. If the agent knows of no gym, the β will be low. If the gym is far away, the δ will be high. The agent will continue their previous activity (a B decision in Eq 2 above). The dynamic described in Eqs 1 & 2 allow us to isolate and measure the effect of information dosing on knowledge about and use of selected resources for health maintenance or promotion activities that were the focus of the CRx intervention. The model does not include other health promotion and maintenance activities occurring at other places, for example going for a walk outside or an informal support group at a home.

## 3.6 CRx ABM submodel: Delivering the HealtheRx in silico

Because our ABM was designed to conduct *in silico* experimentation on the CRx intervention, it includes a component submodel that enables simulated delivery of the HealtheRx to agents. The CRx system's algorithms (described above) used data recorded in each patient's electronic medical record to generate a HealtheRx at each clinical visit. To enable population-level experimentation with the intervention, we implemented these HealtheRx algorithms as a submodel in the CRx ABM and delivered HealtheRxs to agents presenting for healthcare visits. The accuracy of the *in silico* generation of the HealtheRx was compared against 26,558 HealtheRxs

generated *in vivo* during the CRx-2 clinical trial (7/1/16-9/16/16) for residents of the target region. We compared the resources listed on the *in silico* HealtheRx to the *in vivo* HealtheRx based on each agent's sociodemographic characteristics. Validation results for these comparisons were described.

To apply the *in silico* intervention to the synthetic population, agents were assigned health conditions and a preferred language using data from the same 26,558 HealtheRxs. Agent characteristics were statistically matched by gender, age, race and ethnicity to the patients for whom the *in vivo* HealtheRxs had been generated. HealtheRxs were distributed to agents at clinic locations where CRx was deployed *in vivo* during the CRx-1 and CRx-2 clinical trials. [10,11]

### 3.7 Model implementation and validation

A comprehensive description of the model calibration and validation process has been previously published.[39] The Community Rx ABM determined each agent's activity at every hourly time step and moved agents to their corresponding locations. Agent state and location changes were selectively logged, as were the different types of information dosing events. The total number of agents who chose to use a specific resource following the A/B decision process, was also logged. To calibrate and validate the model, we selected resource use in our agent population for 10 specific resources (clinics) that required an A/B decision and calibrated our parameter set where the model output (average resource use) was within a defined range of empirically observed clinic visits. A random forest model was iteratively fitted to model evaluations to characterize the model parameter space against observed empirical data.[39] Table A in S1 Text describes the parameters used in the CRx ABM.

### 4.0 *In silico* experimentation

Experimentation was done using the Extreme-scale Model Exploration with Swift[51] framework on the Midway2 computing cluster at the University of Chicago and the Bebop cluster at Argonne National Laboratory.

### 4.1 Generating endogenous information diffusion networks

We simulated the information diffusion processes within the CRx ABM and generated endogenous information diffusion networks based on hourly snapshots of co-located agents exchanging information based on $\varepsilon_x$ values (described in Section 3.5.2) and the propensity to share information. In these networks, individual agents were network "nodes," and information exchange events between agents were the network "edges." The links in the networks signified the pathways through which information about resources was exchanged. We created an exemplar age-stratified visualization of these networks using Gephi (https://gephi.org) (Fig 6).

### 4.2 Experiments on propensity for information sharing

We ran computational experiments, varying input parameter values that governed the CRx ABM's information diffusion processes. This process helped us analyze how propensity to share information (p-score) affects population level information networks. We parameterized the low, medium and high p-scores into adjustable rates of information exchanges to modify a fundamental generator of the information diffusion network. We show difference in network degree distributions across different levels of p-scores (Fig 7).

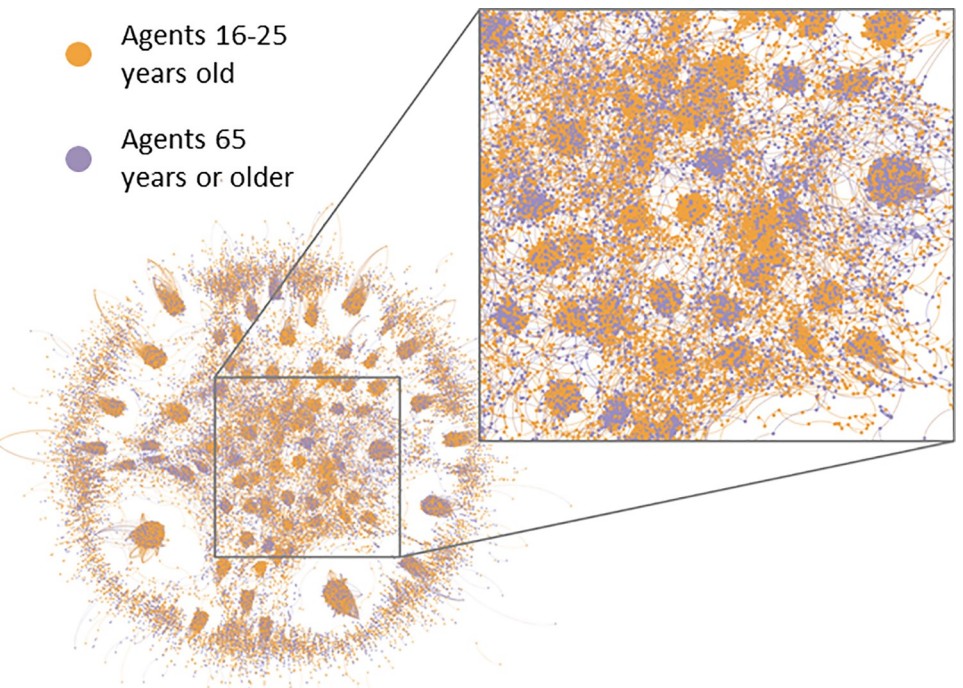

**Fig 6. Visualization of network pathways through which agents, stratified by age (16–25 years = orange, 65 + years = purple) exchange information about resources; Chicago, Illinois 2016–2018.**

## 4.3 Experiment to determine optimal mode of CommunityRx intervention delivery

To estimate the magnitude of the spread of HealtheRx information from social dosing alone, we calculated the ratio of agents who retained HealtheRx information from social dosing exclusively to those who retained HealtheRx information from clinical dosing (using logged simulation data). Using a calibrated parameter set (technical description of model calibration and validation are summarized in Section 3.7 and described in detail in [39]), we used the ABM to run 4 week long simulations, with 15 runs for each of 3 HealtheRx delivery modes (delivered by a physician, nurse, or clinical clerk). We generally observed stable behavior by week 3 of the simulation. We used week 3 outputs to report results (the first two weeks were

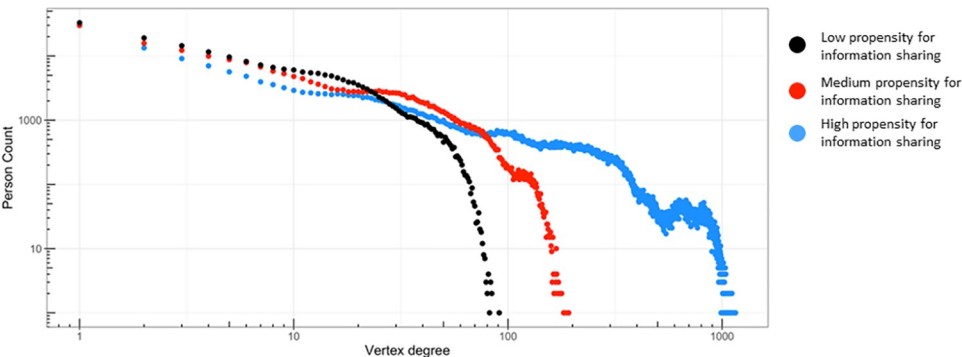

**Fig 7. Differences in network degree distributions (distribution of total incoming and outgoing information pathways) as the rates of information exchanged are adjusted higher, black to red to blue; Chicago, IL 2016–2018.**

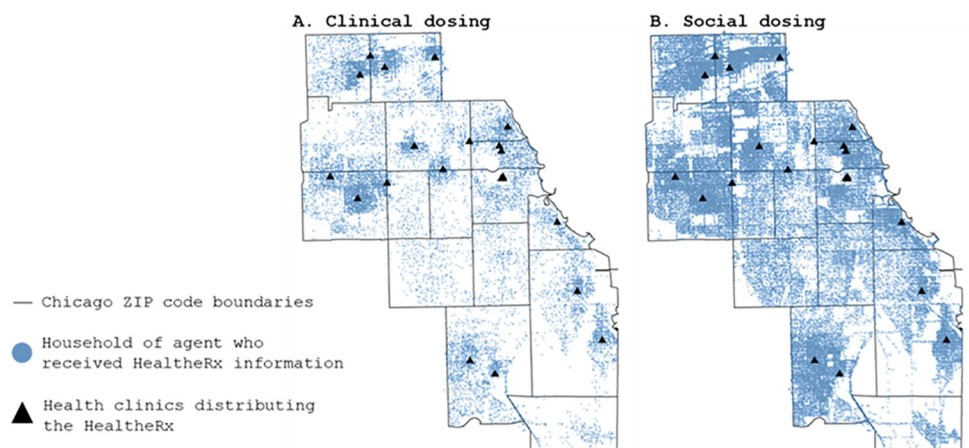

**Fig 8.** Simulation of the geographic spread of community resource information via (A) clinical dosing and (B) social dosing using the CRx agent-based model. Agents who received clinical dosing could also receive social dosing. The base layer of this map was obtained from Stamen Maps available at https://stamen.com/open-source/.

used as the burn-in period for the simulation). Parameter values for the 45 experimental runs are shown in Table C and D in S1 Text. For qualitative comparison of geographic spread of information across the 16 ZIP codes in the CRx ABM, we plotted the home location of agents who retained information from direct clinical versus social dosing during week 3 of the simulation (Fig 8 and Appendix B in S1 Text).

## Results

### Establishing a working model

The mean age of the synthetic population (N = 802,191) was 44 (range: 16–94). The majority of the population was female (56%), non-Hispanic (84%), and African American or Black (59%) (Table B in S1 Text).

Agents, on average, spent the most time engaging in activities during which they could use the following HealtheRx service types: spiritual services, group exercise classes, walking groups, help paying mortgage and rent, and getting fresh fruits and vegetables. The average number of minutes per week agents spent doing activities mapped to these five self-care service types varied by age (Fig 4). For example, compared to agents younger than 30 years old, agents 65 years and older spent, on average, 4.6 times more minutes per week doing activities where they could use spiritual care services.

When we validated the *in silico* delivery of the HealtheRx against 26,558 HealtheRxs delivered *in vivo* during the CRx-2 trial, we found that 91% of community-based resources that were listed on the *in silico* HealtheRxs exactly matched those listed on HealtheRxs generated *in vivo*. The 9% discrepancy can be explained by two factors. *In vivo*, the CRx resource inventory used to generate HealtheRxs was dynamic; places, and the services provided at those places, were routinely updated (e.g., if a place stopped offering a service, it could be deleted from the system). The *in silico* delivery of the HealtheRx used a static community resource database that could not account for changes occurring during the *in vivo* comparison period, 7/1/16-9/16/16. Secondly, *in vivo*, the HealtheRx was limited to 3 pages printed and included an average of 31 resources (range 2–40) whereas the *in silico* HealtheRxs all included 40 resources.

Agent knowledge about resources evolved over time, dependent on an agent's information dosing events. Fig 5 shows the β score dynamics for three agents' knowledge about eight

different resources over time. As expected, resource knowledge increased over time with successive information dosing events (i.e., receiving a HealtheRx, using a resource, or getting information about a resource from another agent), especially those from sources with greater influence (like receiving a HealtheRx and resource use) and decreased when information dosing events did not occur.

By tracking information exchange events that occurred as agents moved through their activity schedules, we generated emerging information diffusion networks. The links in networks signified the pathways through which information about community resources was exchanged. An example diffusion network, generated over the course of a single hour for two age groups—agents 16 to 25 years old and agents 65 or more years old—showed differential clustering by age groups (Fig 6). Information exchange "hubs" ranging from predominantly younger to predominantly older agents emerged.

As we adjusted the rate of propensity for information exchange from low to medium to high, the network degree distributions changed (Fig 7); the number of links (pathways through which information was exchanged) between agents increased. At the highest rate of information exchange, the tail of the distribution extended across three orders of magnitude compared to the lowest rate of information exchanged. Idealized network models have been shown to exhibit exponential,[52] stretched exponential,[53] or scale-free[54] degree distributions. In contrast, these emergent networks, endogenously generated from the local interactions of co-located agents following their individualized schedules, displayed degree distributions with complex structures, which are not as readily fit to simplified functional forms.

### *In silico* clinical versus social dosing experiment

The geographic spread of HealtheRx information via social dosing far exceeded direct clinical dosing alone (Fig 8). Compared to clinically dosed agents (mean = 29,028; range 27,271–30,701), an average of 4.2 times more agents (range 3.9–4.6) retained HealtheRx information from social dosing alone (mean = 123,562; range 118,175–126,658). Table E in S1 Text details the results from each of the 45 simulation runs. The pattern of information spread by agents receiving a HealtheRx at a clinic visit was similar when the HealtheRx was delivered by physicians, nurses or clinical clerks (Appendix B in S1 Text).

## Discussion

CRx is an information-based intervention that connects patients to healthful community resources. In prior studies,[10,11] we discovered that nearly half of patients and clinicians who received the intervention shared information about resources with others, providing evidence of a potentially potent but previously unexplored pathway through which a clinic-based intervention can impact population health. Based on these observations, this study describes how we built a computational laboratory to facilitate the quantification of the population-level impact of CRx. Prior studies of similar information-based interventions have relied on conventional observational and experimental designs.[5] These designs capture the interventions' individual-level impact, but not the population-level effects resulting from the ease with which health-promoting information from the intervention can spread. Information technology is enabling the rapid growth of information-based interventions in public health and health care. [55] To our knowledge, ABM has not yet been used to study the population-level impact of a health information intervention delivered during a clinical encounter.

In contrast to a clinical trial, a computational laboratory is designed to efficiently explore "what-if" questions at large-scale. In the case of CRx, a clinical trial asked "What is the effect of a community resource information intervention on middle age and older adult patients' self-

efficacy for self-care?" We conducted an experiment to address the question: "What is the effect on information diffusion if the intervention is delivered by a physician versus a nurse versus a clerk?" Overcoming experimental capacity of a single clinical trial, the ABM will also allow us to ask other important questions like: "What if the patients who get the intervention are young adults?" Or, "What if the intervention is delivered based on screening versus using a universal approach? What will be the impact on health outcomes and how will the population effects of the intervention be different?"

This study shows how an interdisciplinary team of biomedical and systems scientists collaborated to build a computational laboratory. We have demonstrated that it is realistic enough for its purpose, by observing patterns that replicated intuitive or previously-known results. As expected, we demonstrated that synthetic agents exhibited diverse behavioral characteristics, reflective of the known geographic population, and that the synthetic intervention could be delivered to these agents *in silico*. We also showed that agents' resource knowledge evolved as they were exposed to resource information and evolved differently depending on the information source. We illustrated the emergence of information diffusion networks and how the degree distributions of these networks changed as the values of key parameters like the propensity of information sharing were varied. Lastly, we demonstrated the computational feasibility of using highly granular activity data, an important feature of a study seeking ultimately to uncover when, where and how information spreads between people in a population. Combined with prior validation work, these results help to establish the model's readiness for hypothesis-testing experimentation.[38,39]

To illustrate the potential of the ABM as a complement to clinical methods, we also conducted an *in silico* experiment to examine how specific resource information disseminates from an individual exposed to the CRx intervention in a clinical setting ("clinical dose") to other individuals in the population through social dynamics ("social dose"). Through 45 experimental runs simulating mixing behaviors in a geographic population, this study finds that personalized community resource information delivered to a patient during a healthcare visit spreads to others in the community. Complementing the three-month trial's observation that the total number of people reached by HealtheRx information was at least double the number of middle-aged and older patients in the intervention group [17], the ABM *in silico* experiment estimates this ratio to be approximately 4-fold, when including all people ages 16 and older in the geographic population. We also find that the mode of clinical delivery (physician, nurse, clerk) likely has little relative impact on the magnitude or dynamics of spread, suggesting that clinical implementation of community resource referral solutions might focus less on who delivers the intervention and more on optimizing patient and clinician access to it. These empirical and simulated observations give life to the concept of patient social dosing, a previously overlooked mechanism, and potential force multiplier through which community resource referral and other health-promoting information interventions may deliver value beyond the patient who receives direct intervention during a healthcare visit. To our knowledge, this is the first study to examine the population-level diffusion dynamics of a clinical information intervention.

Most ABMs are built using data from extant literature, secondary datasets and expert informants.[42] In 2010, Morell and colleagues proposed an integrated approach where data and insights generated through observational evaluations are used to drive building and parameterization of ABMs.[21] Public health researchers subsequently voiced support for this complementary approach.[19,20] Building on and extending these ideas, the CRx ABM was constructed in tandem with both observational and experimental studies and was parameterized using data from these studies. Building the ABM to simulate agent-level information sharing, a dynamic observed in CRx trials, revealed a need to assign a propensity for information

sharing during agent activities. Because we had not anticipated this data need at the outset of the trial, we generated this parameter set by surveying expert informants. The addition of primary data collection to estimate the propensity for information-sharing during daily activities would increase the value of population-based time use surveys, like ATUS, to *in silico* experimentation with information-based interventions.

In parallel with ABM building, iteration and experimentation, we continue to operate clinical observational and experimental trials to deepen our understanding of how information about resources spreads and the impact of the resource information on a theory-driven set of health and healthcare outcomes.[56,57] These empirical data from clinical trials will be used to further parameterize and improve the ABM over time, which, in turn, will enable more advanced *in silico* experimentation. For instance, the CRx ABM can be used to compare community resource use in relation to varying rates of information spread and network size and other characteristics to inform assessments and decision-making regarding resource allocation. These, and other experiments, are a focus of future work.

Although ABM-based experimentation enables rapid iteration of community-wide trials under varying conditions that would be infeasible to conduct in the real world, there are limitations. The CRx ABM is driven by a decision-making model that is a simplified representation of factors influencing how people decide to use health-promoting community resources. For example, we only account for face-to-face information sharing, not other modalities like social media. The validity of ABM-based experimentation is dependent on the salient factors captured by the underlying behavioral models. Some agent characteristics that are likely dynamic in the real-world are static in this first iteration of the model (e.g., agent activation). Modeling behavior change is an active area of systems science research;[58] we expect that new discoveries will inform iterations of our model over time. Also, the CommunityRx ABM only accounts for knowledge about and use of local health-related resources that could be listed on the HealtheRx. Health maintenance activities could occur outside of these resources (e.g., walking in the neighborhood, an informal support group at someone's home). We also have a limited understanding of which and how many resources a person needs to maintain health, how this information is retained in memory and how these factors vary among individuals. Lastly, the propensity for information sharing during specific activities is an important variable in our ABM, but our estimates may be limited by use of expert informant data. This limitation reveals a data need that will be important to future computational studies of information-based interventions and presents an exciting opportunity for new empirical research. Although these are important limitations, the *in silico* laboratory is flexible; we can iterate the decision model in tandem with emergent empirical data and advances in the field more generally. Thus, while the focus of the CRx ABM is on simulating the CRx intervention, the process of model building and computational experimentation presented is generalizable to other large-scale ABMs, for example those modeling information diffusion processes. However, generalizability may be limited because the model was validated only against the data that were used to inform the model building.

We have demonstrated the process and feasibility of integrating clinical trials and systems science methods to build a flexible laboratory for studying the population health impact of an information-based intervention delivered to individuals at the point of medical care. By using open source tools and sharing our methods,[38,39] we aim to build trust in our approach, prompt feedback from peers, and enable others to use, iterate and learn from our model. In addition to advancing knowledge specific to understanding and valuing the impact of CRx, this work serves to advance knowledge and testing of a fast-growing range of information-based health interventions being developed for delivery to promote patient and population health.

## Supporting information

**S1 Text.** Table A. Agent-Based Model Parameters, Description, Values and Sources of Values. Table B. Demographic Characteristics of the Synthetic Population (N = 802,191) Generated Using Data From the Synthetic Populations and Ecosystems of the World; Chicago, IL 2016–2018. Appendix A: Summary Model Description following the Overview, Design concepts and Details (ODD) Protocol (Grimm, 2020). Table C. Common parameter values and ranges for direct versus social dosing experiment runs. Table D. Varied parameter settings experiment runs. Table E. Direct versus Social Dosing Experiment Results. Appendix B. Geographic Spread by Dosing Source.
(DOCX)

## Acknowledgments

The authors (N.C., C.K., C.M.M., J.O.) would like to acknowledge facilities and material resources support provided by the U.S. Department of Energy Office of Science to Argonne National Laboratory under Contract No. DE-AC02-06CH11357.

## Author Contributions

**Conceptualization:** Stacy Tessler Lindau, Jennifer A. Makelarski, Chaitanya Kaligotla, Emily M. Abramsohn, David G. Beiser, Chiahung Chou, Nicholson Collier, Elbert S. Huang, Charles M. Macal, Jonathan Ozik, Elizabeth L. Tung.

**Data curation:** Stacy Tessler Lindau, Jennifer A. Makelarski, Chaitanya Kaligotla, Nicholson Collier, Charles M. Macal, Jonathan Ozik.

**Formal analysis:** Stacy Tessler Lindau, Jennifer A. Makelarski, Chaitanya Kaligotla, Nicholson Collier, Charles M. Macal, Jonathan Ozik.

**Funding acquisition:** Stacy Tessler Lindau, Elbert S. Huang.

**Investigation:** Stacy Tessler Lindau, Jennifer A. Makelarski, Chaitanya Kaligotla, Emily M. Abramsohn, David G. Beiser, Chiahung Chou, Nicholson Collier, Elbert S. Huang, Charles M. Macal, Jonathan Ozik, Elizabeth L. Tung.

**Methodology:** Stacy Tessler Lindau, Jennifer A. Makelarski, Chaitanya Kaligotla, Emily M. Abramsohn, David G. Beiser, Chiahung Chou, Nicholson Collier, Elbert S. Huang, Charles M. Macal, Jonathan Ozik, Elizabeth L. Tung.

**Project administration:** Stacy Tessler Lindau, Jennifer A. Makelarski, Emily M. Abramsohn.

**Resources:** Stacy Tessler Lindau.

**Software:** Chaitanya Kaligotla, Nicholson Collier, Jonathan Ozik.

**Supervision:** Stacy Tessler Lindau, Jennifer A. Makelarski.

**Validation:** Stacy Tessler Lindau, Jennifer A. Makelarski, Chaitanya Kaligotla, Emily M. Abramsohn, David G. Beiser, Chiahung Chou, Nicholson Collier, Elbert S. Huang, Charles M. Macal, Jonathan Ozik, Elizabeth L. Tung.

**Visualization:** Stacy Tessler Lindau, Jennifer A. Makelarski, Chaitanya Kaligotla, Jonathan Ozik.

**Writing – original draft:** Stacy Tessler Lindau, Jennifer A. Makelarski, Emily M. Abramsohn, Charles M. Macal, Jonathan Ozik.

**Writing – review & editing:** Stacy Tessler Lindau, Jennifer A. Makelarski, Chaitanya Kaligotla, Emily M. Abramsohn, David G. Beiser, Chiahung Chou, Nicholson Collier, Elbert S. Huang, Charles M. Macal, Jonathan Ozik, Elizabeth L. Tung.

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
