## [Decision Letter · Decision Letter 0]

6 Apr 2021

Dear Dr. Lindau,

Thank you very much for submitting your manuscript "Building an agent-based model to study the population-level health impact of CommunityRx, a community resource referral intervention" for consideration at PLOS Computational Biology.

As with all papers reviewed by the journal, your manuscript was reviewed by members of the editorial board and by several independent reviewers. In light of the reviews (below this email), we would like to invite the resubmission of a significantly-revised version that takes into account the reviewers' comments.

We cannot make any decision about publication until we have seen the revised manuscript and your response to the reviewers' comments. Your revised manuscript is also likely to be sent to reviewers for further evaluation.

Sincerely,

Benjamin Muir Althouse

Associate Editor

PLOS Computational Biology

Nina Fefferman

Deputy Editor

PLOS Computational Biology

Reviewer's Responses to Questions

**Comments to the Authors:**

Reviewer #1: This article describes the CommunityRX agent-based model (henceforth referred to as the CRX-model) aimed at doing in silico experimentation on an in vivo intervention called CommunityRX. In this intervention information of local health resources is shared with individuals during their touch point with the health system. As individuals had indicated they share such information, the CRX-model is aimed at capturing evidence of the propagating systemic impact of such an intervention (beyond the cohort directly exposed). The manuscript is well written and the method of using ABM to explore both the size and mechanism of the systemic impact has both merit, can provide invaluable for decision support, translates far beyond the setting in which it is current applied, and as such would be of interest to the journal audience.

The self-reported goal of the manuscript is “to describe the interdisciplinary processes and data sources used to develop an ABM to study an information-based health intervention“ (L164/165), yet I believe it falls short of this aim. The model description provided is too high level to fully grasp model functioning or its details and consequently for many of the details the two previous studies on the model are cited [30/31]. If model description is indeed the goal, there are standards such as ODD (Grimm et al. 2006,2010,2020) that would greatly improve the manuscript structure and would provide guidance as to how to accurately do so. The more fundamental question is if model description should in fact be the desired purpose of the manuscript. I would deem a manuscript that only provide a model description as outside the scope of PLOS Computational Biology, and potentially suited for the realm of the contextual health domain. What would be if interest to the journal audience would be a detailed description of how to build this type of ABM (focused on the method of model building), or an illustration of how such an ABM could provide insight trough in silico experimentation. While aligning with either of these aims would require mayor revisions there is merit in doing so.

Regardless of the revision path chosen, a more detailed model description of the model will be required (either as part of the Supplementary information, or in the main text), and the following comments would support these efforts.

1. Provide an overview of model behaviors, prior to specifying the details (moving section implementing the ABM to the start of the methods section would be a good start).

2. Include a pseudo code description of the model behaviors to that overview section

3. Provide the characteristics of the population (individual characteristics and their distribution)

(note that the above should follow naturally form adopting ODD as the standard for model description)

4. The schedule assignment is random (L230 – 234), which has a large randomizing effect on the networks that will be created. While it is debatable if this is realistic, the implications of this modeling decision should be explored, or be made explicit at the least.

5. The mapping of ATUS activities and CRX resources should be clarified, currently it is not clear from the text how exactly this works, and what the scope of activities are. More details are needed.

6. Clarify the information sharing process in more detail (see comment below for details)

An general modeling concern relates to the specification of the information sharing mechanism in the CRX-model. The authors signal the awareness of the notion that information transfer to consist of three steps (sending information, transmission, and reception) by stating the steps required in the vivo intervention result in administration of the intervention (L122-131). In each propagating process there will be the action of sending, the medium transmitting that signal, and a reception processes by which the recipient processes the signal (and updates their state or behavior). While these are touched upon for the initial transfer during the in vivo intervention, for consequent information sharing among peers this is not the case. In fact you based the notion of secondary information transfer solely on the account that a respondent have indicated they shared information, without evidence that behavior is changed as a result of this sharing the presumed systemic effect is in fact speculative. As far as I can tell form the manuscript the steps following sharing are (transfer and reception) are assumed to occur as a given. What is more is seems a if sharing occurs in a broadcasting manner, touching all who are collocated. As the secondary spread is the sole driver of the presumed systemic impact a more detailed elaboration of such assumptions is required for this model.

Going back to the revision paths; If the goal will be to present the process of model building in detail there are concerns relating to the overlap with earlier manuscripts, specifically [31]. It will need to be made evident what is novel in the current submission. Alternatively, If the goal will be to present the use of this model for decision support or even learning about the mechanisms of secondary spread the outcomes will need to be refined. In fact apart from the degree distribution the results presented in the current manuscript are in fact input data rather than model outcomes. What is more, the degree distribution of the spread network might not be the most informative outcome. Comparisons of community resource use under different rates of spread, network size (both in terms of edges and vertices) might allow you to drive home points relating to reach and impact on behavior much better.

Lastly, as a minor comment I would add that the clarity of figures would benefit from higher resolution images.

Reviewer #2: The authors have submitted a very interesting manuscript about an advanced topic. The integration of (clinical) trials and systems science methods for studying population health impacts of (medical) interventions is a combination that will become more common in near future. In their manuscript they clearly describe an example of the process of this integration which can guide others.

However, I found some issues where the manuscript needs improvement before it meets the expectations of a good model of this integration process.

First, a manuscript describing a complex agent-based model should follow generally accepted reporting guidelines like the ODD Protocol for Describing Agent-Based and Other Simulation Models (DOI: 10.18564/jasss.4259).

Comments in detail:

line 135-136: "tens of thousands"

line 221: Where were the characteristics described? Was the synthetic population static or dynamic?

line 231: The schedules were randomly assigned. Did you include differences between weekdays, weekly and seasonal influences on the daily schedules? If not, please discuss the reasons.

Generating agents' decisions:

- You included a very strong restriction in the model by forbidding agents to conduct activities matching CommunityRx resources if they did not use such a resource (i.e. not doing the activity without using a resource). Activities like going for a walk or bible studies might be possible without using a CommunityRx resource in reality. I think this strong restriction of the model might lead to a significant overestimation of the impact of providing information and of information sharing. Please discuss.

- Information about a resource was modeled to have an enhancing effect only - not a negative one, which can be also possible in reality. Please discuss.

Generating agents' information sharing behaviors and knowledge evolution:

- Did you consider a kind of "burn-in" period? How long did it take until no effects of the arbitrary initial values were observable any more?

- The necessary interaction between the donor and receiver of information about resources is not sufficiently described, e.g. how was determined, which information (information about which resources) was shared? Only the propensity of the receiver is mentioned, but no properties of the donor.

- line 351: Where were the validation results described?

Results:

- Which simulated time period was used to get the results?

- The model was validated only against data that was used to inform the model. This is not really an external validation and may limit the generalizability of the results. Please discuss.

Discussion:

- lines 483-485: These were design descisions, i.e. this behavior of the model was expected. Something would have been wrong in the implementation if the model did not show this behavior. Please clarify.

Reviewer #3: Review uploaded as an attachment.

Reviewer #4: The following are my general comments of the article

The article shows a coherent story of the study, the authors have taken the time to explicitly explained the whole concept of Agent Based Modeling and its associated procedures. The arrangements were made in such a way that, it predominantly centered on language understandable by all thus non-technical interpretations were offered.

In as much as the reviewer believes detailed procedures and well explained methodology and implementation understandable by ordinary people with interest in science, the author also suggests the following below for the authors to reconsider in providing further explanations to make the concepts much clearer.

Report on Reviewed article

COMMENT: Line 133 The authors should consider deleting line 133 sentence as it doesn’t add up to the sentence. It can be placed at where funding information is required to be place

133 With initial funding from a Center for Medicare and Medicaid Innovation award (3/2013-5/2015,

134 “CRx-1”),

COMMENT: Line 135 I believe the authors were referring to tens of thousands instead of "thens" of thousands

135 30 month period. All patients living in the target geography were eligible, including thens of

136 thousands of people with cancer, cardiovascular disease, pulmonary disease, obesity and other

COMMENT: Line 137 Authors should include the unit of measurement of age, “18-93 years”

137 common chronic conditions. In a telephone survey of 458 HealtheRx recipients ages 18-93,

COMMENT: Line 197 This statement should be appropriately recapture and the link should be inserted as part of the appendix. Moreover, authors should clearly show which part is in a bracket and which is not, the end of the link has a close bracket but the beginning of the bracket is not seen.

197 toolkits. The model code and the workflow code used to implement the parameter space

198 characterization experiments are publicly available at https://github.com/jozik/community-rx).

COMMENT: Line 250. What informed this categorization, any special reasons? as why not other age groups different from what you have chosen?

250 was computed within each of four age groups (<31, 31-45, 46-65 and 65+ years), weighted by

COMMENT: Line 276: Can authors throw more light as to how (inherent inertia) this was captured or measured? As it could be a source of uncertainty in the model itself? Was it also derived from Skolasky et al?

276 iii) the inherent inertia that they needed to overcome for using that service (�j).

COMMENT: Line 305/308: In equation presentation, variables/parameters, constants are explained just below the equations, in equation 2, since all variables have been previously explained, could authors given meaning to the value, �x, where ∈?. Moreover, �x could have been any value, why the chosen values on line 308?

305: of resource information was associated with a value, �x, where ∈

308: used to produce Figure 5, the following values were used = {0.05, 0.15, 0.25, 0.2, 0.9, 1}.

NOTE (Not necessarily a required consideration, but a simple explanation could help, on below general comments)

1. It is widespread believed that, the alternative version of ABM could be made in differential equations since its well suited in dynamical systems of how agents behave (though the debate has not settled yet within the mathematical science fraternity), could the authors consider explaining why ABM is well suited for this study but not any of the Complex Adaptive Systems Modeling techniques (Complex Network Modeling Level, Exploratory Agent-based Modeling Level, Validated agent-based modeling and Descriptive Agent-based Modeling )?

2. The GitHub link provided by the author: Authors should consider adding in-depth comments to the scripts provided in the GitHub in order for users who might "clone the scripts" be able to understand what the lines of codes were meant for, for the sake of reproducibility of work which is the cornerstone of today's scientific study dissemination.

**Have all data underlying the figures and results presented in the manuscript been provided?**

Reviewer #1: Yes

Reviewer #2: None

Reviewer #3: Yes

Reviewer #4: Yes

PLOS authors have the option to publish the peer review history of their article (what does this mean?). If published, this will include your full peer review and any attached files.

Reviewer #1: No

Reviewer #2: **Yes: **Veit Zoche-Golob

Reviewer #3: **Yes: **A K Chattopadhyay

Reviewer #4: No
---

## [Decision Letter · Decision Letter 1]

4 Aug 2021

Dear Dr. Lindau,

Thank you very much for submitting your manuscript "Building and experimenting with an agent-based model to study the population-level impact of CommunityRx, a clinic-based community resource referral intervention" for consideration at PLOS Computational Biology.

As with all papers reviewed by the journal, your manuscript was reviewed by members of the editorial board and by several independent reviewers. In light of the reviews (below this email), we would like to invite the resubmission of a significantly-revised version that takes into account the reviewers' comments.

We cannot make any decision about publication until we have seen the revised manuscript and your response to the reviewers' comments. Your revised manuscript is also likely to be sent to reviewers for further evaluation.

Sincerely,

Benjamin Muir Althouse

Associate Editor

PLOS Computational Biology

Nina Fefferman

Deputy Editor

PLOS Computational Biology

Reviewer's Responses to Questions

**Comments to the Authors:**

Reviewer #2: The revised version of the manuscript is much clearer than the previous version and I appreciate the authors' efforts to follow the reviewers' suggestions. However, the manuscript would still benefit from more comprehensive presentation and interpretation/discussion of the in silico experiments' results. And as this paper is intended to serve as an inspiring example, it is crucial to discuss the generalizability of the results of these experiments.

Unfortunately, some of my previous comments/ questions were not sufficiently answered.

1) Methods Sections 3.5.2 and 4.3, Results: Were the outputs of the first two weeks discarded which were influenced by the initialialization of the agents ("burn-in period")? Were only the outputs of following 4 (or 2?) simulated weeks considered for generating the results (e.g. to calculate the average number of minutes per week spent doing each activity? Please clarify.

2a) Methods Section 3.5.3 Agent information sharing behaviour: Did all agents that were at the same place at the same time share their information? (If 10 persons were at a place at the same time, would any of them receive information about all resources known by all other 9 persons? - Or was some kind of random matching of subsets of present persons performed?

2b) lines 511-512: As far as I understand, only the characteristics of the receiver of information were included (p-score), but not of the donor of information. Then "receiving" should be replaced by "providing" or "giving" in this sentence.

3) The manuscript still lacks a discussion of the strong restriction that an agent could only conduct an activity that includes "physical and mental health maintenance" if she/he decided to use a rescource (Decision A), otherwise she/he would continue doing the previous activity in their schedule (Decision B) (lines 531-535). As I already mentioned in my review of the previous manuscript version, this restriction of the model might lead to a significant overestimation of the impact of providing information and of information sharing. Therefore, this model restriction, the reasons for its inclusion and its potential influence on the model outcome need to be discussed.

4) My previous comment "The model was validated only against data that was used to inform the model. This is not really an external validation and may limit the generalizability of the results. Please discuss." was not considered appropriately in the current version of your manuscript. Although Methods Section 3.7 clarifies the calibration and validation process, in the discussion, e.g. lines 747-748, 750-768 (and the Abstract, l. 69-71) any statement on the limited generalizability of the results of the experiment is missing. Instead, the simulation results are presented in a way as if they were generally valid.

Please make your own opinion ("While we calibrate and validate our model against empirical data, we do not claim model generalization beyond recreating the CommunityRx simulation in silico. Our general process of model building and the use of computational laboratories, however, is generalizable.") absolutely clear in the Discussion section and the Abstract.

Additionally, I found some minor issues:

line 300: "HealtheRxs"

line 318: "exchanging"

Supplement Table 1: Full citation of "Garibay, 2011" is missing (or is it Garibay, 2014?)

Reviewer #4: Review is uploaded

**Have the authors made all data and (if applicable) computational code underlying the findings in their manuscript fully available?**

Reviewer #2: Yes

Reviewer #4: Yes

PLOS authors have the option to publish the peer review history of their article (what does this mean?). If published, this will include your full peer review and any attached files.

Reviewer #2: **Yes: **Veit Zoche-Golob

Reviewer #4: **Yes: **Emmanuel de-Graft Johnson Owusu-Ansah
---

## [Decision Letter · Decision Letter 2]

23 Sep 2021

Dear Dr. Lindau,

We are pleased to inform you that your manuscript 'Building and experimenting with an agent-based model to study the population-level impact of CommunityRx, a clinic-based community resource referral intervention' has been provisionally accepted for publication in PLOS Computational Biology.

Best regards,

Benjamin Muir Althouse

Associate Editor

PLOS Computational Biology

Nina Fefferman

Deputy Editor

PLOS Computational Biology

Reviewer's Responses to Questions

**Comments to the Authors:**

Reviewer #2: Thank you for the opportunity to review your interesting manuscript and answering my questions in detail. I have no further comments.

**Have the authors made all data and (if applicable) computational code underlying the findings in their manuscript fully available?**

Reviewer #2: Yes

PLOS authors have the option to publish the peer review history of their article (what does this mean?). If published, this will include your full peer review and any attached files.

Reviewer #2: **Yes: **Veit Zoche-Golob

---

## [Editor Report · Acceptance letter]

15 Oct 2021

PCOMPBIOL-D-20-02061R2 

Building and experimenting with an agent-based model to study the population-level impact of CommunityRx, a clinic-based community resource referral intervention

Dear Dr Lindau,

I am pleased to inform you that your manuscript has been formally accepted for publication in PLOS Computational Biology. Your manuscript is now with our production department and you will be notified of the publication date in due course.

With kind regards,

Andrea Szabo
